# Irisin Attenuates Muscle Impairment during Bed Rest through Muscle-Adipose Tissue Crosstalk

**DOI:** 10.3390/biology11070999

**Published:** 2022-06-30

**Authors:** Andrea D’Amuri, Juana Maria Sanz, Stefano Lazzer, Rado Pišot, Bostjan Šimunič, Gianni Biolo, Giovanni Zuliani, Mladen Gasparini, Marco Narici, Bruno Grassi, Carlo Reggiani, Edoardo Dalla Nora, Angelina Passaro

**Affiliations:** 1Medical Department, University Hospital of Ferrara Arcispedale Sant’Anna, Via A. Moro 8, I-44124 Ferrara, Italy; dmrndr@unife.it (A.D.); giovanni.zuliani@unife.it (G.Z.); 2Department of Chemical, Pharmaceutical and Agricultural Sciences, University of Ferrara, Via Luigi Borsari 46, I-44121 Ferrara, Italy; juana.sanz@unife.it; 3Department of Medicine, University of Udine, Piazzale M. Kolbe 4, I-33100 Udine, Italy; stefano.lazzer@uniud.it (S.L.); bruno.grassi@uniud.it (B.G.); 4Institute for Kinesiology Research, Science and Research Centre Koper, Garibaldijeva 1, SI-6000 Koper, Slovenia; rado.pisot@zrs-kp.si (R.P.); bostjan.simunic@zrs-kp.si (B.Š.); carlo.reggiani@unipd.it (C.R.); 5Department of Medicine, Surgery and Health Sciences, University of Trieste, Strada di Fiume, 447, I-340149 Trieste, Italy; biolo@units.it; 6Department of Translational Medicine, University of Ferrara, Via Luigi Borsari, 46, I-44121 Ferrara, Italy; 7Department of Vascular Surgery, Izola General Hospital, Polje 40, SI-6310 Izola-Isola, Slovenia; mladen.gasparini@sb-izola.si; 8Department of Biomedical Sciences, University of Padua, Via Marzolo 3, I-35131 Padua, Italy; marco.narici@unipd.it; 9Research and Innovation Section, University Hospital of Ferrara Arcispedale Sant’Anna, Via A. Moro 8, I-44124 Ferrara, Italy

**Keywords:** inactivity, FNDC5 gene expression, myokines, sarcopenia, muscle fiber

## Abstract

**Simple Summary:**

Irisin is a known myokine secreted mainly by the muscle that is produced after physical activity. It induces browning in the adipose tissue with a consequent increase in mitochondrial oxidation of lipids and reduction of insulin resistance; thus, it has been hypothesized that irisin was the molecule mediating most of the beneficial effects related to exercise on adipose tissue and consequently on the whole organism. In our study we observed that extreme physical inactivity induces the loss of muscle mass and function, and an increase in the body adipose tissue as expected. However, of note, circulating irisin levels were increased secondary to enhanced irisin synthesis mainly from adipose tissue rather than muscle. In addition, subjects who produced more irisin had reduced muscle impairment. Therefore, our hypothesis is that there is negative feedback within the muscle-adipose tissue crosstalk, specifically not only does the muscle influence the adipose tissue through irisin during exercise, but also the adipose tissue protects the muscle during inactivity.

**Abstract:**

The detrimental effect of physical inactivity on muscle characteristics are well known. Irisin, an exercise-induced myokine cleaved from membrane protein fibronectin type III domain-containing protein-5 (FNDC5), mediates at least partially the metabolic benefits of exercise. This study aimed to assess the interplay between prolonged inactivity, circulating irisin, muscle performance, muscle fibers characteristics, as well as the FNDC5 gene expression (FNDC5ge) in muscle and adipose tissue among healthy subjects. Twenty-three healthy volunteers were tested before and after 14 days of Bed Rest, (BR). Post-BR circulating levels of irisin significantly increased, whereas body composition, muscle performance, and muscle fiber characteristics deteriorated. Among the subjects achieving the highest post-BR increase of irisin, the lowest reduction in maximal voluntary contraction and specific force of Fiber Slow/1, the highest increase of FNDC5ge in adipose tissue, and no variation of FNDC5ge in skeletal muscle were recorded. Subjects who had the highest FNDC5ge in adipose tissue but not in muscle tissue showed the highest circulating irisin levels and could better withstand the harmful effect of BR.

## 1. Introduction

During muscle contraction, myocytes act as a secretory organ releasing myokines that work in an autocrine, paracrine, and endocrine manner. Specifically, irisin is a myokine [1] regulated by peroxisome proliferator-activated receptor (PPAR)-γ coactivator-1α (PGC-1α) and cleaved from membrane protein fibronectin type III domain-containing protein 5 (FNDC5). Irisin is a short-life molecule with a high rate of degradation [2,3], involved in the regulation of glyco-lipidic metabolism [4,5,6]. Furthermore, it increases cortical bone mass [7] and protects against muscle mass decline in animals [8]. In this perspective, irisin could mediate some of the health benefits associated with exercise. In fact, it triggers downstream AMPK activation [9], a cornerstone of the exercise-induced molecules that regulates cellular metabolism [10]. It is still debated what type of exercise stimulates irisin production. In fact, Boström et al. originally described irisin release in healthy men following a 10-week period of moderate intensity endurance exercise [1]; this observation was also confirmed during high and low intensity exercise [11], while others reported no associations with endurance or resistance training [12,13,14,15]. These conflicting results lead to questioning the existence of circulating irisin [16] until Jedrychowski et al. unequivocally demonstrated its presence in humans and its increase following exercise [17]. Although a correlation between irisin and exercise was proved, the ideal enhancing stimulus of irisin release is still unclear. Chronic responses after training bouts showed controversial results with either a modest increase [1,18,19] or no effects [12,13,20,21] in plasma irisin. However, the comparison between studies might be unreliable due to different training protocols in terms of type (aerobic vs. strength training) intensity (low, moderate, or high), duration and frequency of exercise. In addition, timing of blood sampling was extremely variable after exercise [12,14]. Conversely, prior work showed an irisin response to acute bouts of exercise [2,9,14,18,22], immediately after the exercise, depending on the intensity [9,11]. Interestingly, when acute exercise is repeated constantly during many weeks, becoming a chronic stimulus, it fails to induce a detectable irisin increase [9]. Among the unknowns related to irisin release, the relationship with age is also debated [2,23,24]. Finally, there are no studies that have explored the modulation of irisin during physical inactivity. We have previously observed that the response to Bed Rest (BR), a model of extreme physical inactivity, is highly variable among individuals in terms of body composition, muscle characteristics, and properties of muscle fibers [25].

To understand whether irisin is associated with the biological consequences of BR, we assessed irisin plasma concentration, body composition, muscle function, size and performance of single muscle fibers, as well as the FNDC5 expression on muscle and adipose tissue before and after an experimental 14-day BR in a cohort of healthy volunteers.

## 2. Materials and Methods

### 2.1. Participants

Healthy volunteers underwent horizontal BR for 14 days (BR14) in standard air-conditioned hospital rooms of the Orthopedic Hospital of Valdoltra (Slovenia) under 24-h surveillance and medical care. During BR, all subjects received an individually controlled normo-caloric diet: resting energy expenditure was multiplied by factor 1.2, with caloric content distributed as follows: 60% carbohydrates, 25% fat, and 15% proteins. Participants performed daily activities in bed (i.e., communicate, watch television and listen to radio, read, use computer, and to receive visitors [26]). To evaluate the effect of cognitive stimuli on BR consequences, a subgroup of eight randomly selected older adults underwent 45 min daily of Computerized Cognitive Training (CCT-Older) by navigating through virtual mazes with the use of a joystick and computer during the BR.

Exclusion criteria were: smoking; regular alcohol consumption; ferromagnetic implants; history of deep vein thrombosis with D-dimer levels at enrollment greater than 500 μg L^−1^; acute or chronic skeletal, neuromuscular, metabolic and cardiovascular disease conditions; pulmonary embolism.

### 2.2. Biologic Samples and Measurements

All samples were obtained at the baseline data collection (BDC) and at the end of BR14. Blood samples were collected after an overnight fasting and centrifuged. Aliquots were stored at −80 °C. Plasma irisin was measured by ELISA (Adipogen). The coefficient of variation was <7% and <10% for intra-assay.

### 2.3. Anthropometric Characteristics and Body Composition

Body mass was measured to the nearest 0.1 kg with a manual weighing scale (Seca 709, Hamburg, Germany). Stature was measured to the nearest 0.5 cm on a standardized wall-mounted height board. Body mass index was calculated (BMI = Body weight (kg)/Stature (m) × Stature (m)). Body composition was measured using bio-impedance with a tetra-polar impedance-meter (BIA101, Akern, Florence, Italy) according to manufacturer’s instructions (Lukaski, Johnson, Bolonchuk and Lykken, 1985), all measures were collected by trained staff members, after eight hours fasting in horizontal position.

### 2.4. Muscle Characteristic

Quadriceps femoris muscle volume (QMV) of the right leg was measured from turbo spin-echo, T1-weighted, magnetic resonance images (MRI) obtained with a 1.5 T Magnetom Avanto device (Siemens Medical Solution, Erlangen, Germany) as previously described [27]. The biomechanical parameters of the maximal explosive efforts (MEP) were studied by means of an Explosive Ergometer (EXER) [28]. Furthermore, specific MEP = MEP/QMV was also calculated. Maximal voluntary isometric contractions (MVC) of the right lower limb knee extensors were performed on a special chair as already described [27]. Furthermore, specific MVC = MVC/QMV was also calculated.

### 2.5. Muscle Fiber Analysis

Single muscle fiber analysis was performed [26,27] on biopsy samples obtained from the mid-region of the left vastus lateralis muscle. Biopsy was carried out after anesthesia of the skin, subcutaneous fat tissue, and muscle fascia with 2 mL of lidocaine (2%). A small incision was then made to penetrate skin and fascia, and the tissue sample was harvested with a purpose-built rongeur (Zepf Instruments, Tuttlingen, Germany). A fragment of the sample, used for single fiber analysis, was quickly stored in skinning solution with 50% glycerol at −20 °C, while another fragment was frozen in isopentane cooled with liquid nitrogen. Further details of the protocol and of the experimental setup were reported elsewhere [29]. The fiber segments were gently elongated in the relaxing solution to 120% of the slack length, which corresponded to a sarcomere length of 2.685 ± 0.011 μm (mean ± SE, *n* = 710). The segments were then transferred to the pre-activating solution for at least 1 min and finally, maximally activated by immersion in the activating solution. During maximal activation, isometric force (F_0_) was measured in several consecutive isometric contractions and unloaded shortening velocity (V_0_) was determined according to the slack test procedure. To this end, five instantaneous length changes (<1 ms) were imposed with amplitudes ranging from 5 to 15% of the resting length. Unloaded shortening velocity was obtained from the slope of the linear regression between the time required to take up the slack and the amount of shortening imposed and was expressed in fiber length per second. Cross-sectional area (CSA) was calculated from the measurements of three fiber diameters, assuming a circular shape of the fiber section, while the fiber was immersed in relaxing solution. Furthermore, specific force (P_0_) = F_0_/CSA was also calculated. The composition in Myosin Heavy Chains (MyHC) isoforms of each fiber segment was determined on 8% polyacrylamide slab gels after denaturation in SDS (SDS-PAGE) as described by Doria et al. [29]. Gels were silver stained, and three bands were separated in the region of 200 kDa, corresponding (in order of migration from the fastest to the slowest) to MyHC-1, MyHC-2 (MyHC-2A and MyHC-2X). The same protocol was followed to separate MyHC isoforms in the frozen fragment of the biopsy, with Coomassie Blue staining of the gels. The relative proportions of MyHC isoforms were obtained from the measurements of the brightness area product (BAP, i.e., the product of the area of the band by the average rightness subtracted local background after black–white inversion) after scanning the gels to an accuracy of 600 dpi. For each sample, the electrophoretic separation and the densitometry measurements were repeated three times

### 2.6. Adipose Tissue Gene Expression

Adipose tissue samples were obtained from each participant from the gluteus. Briefly, the subject was instructed to hold in tension the muscles, so that the muscle and the fat pad were clearly recognizable. A fold from the upper outer quadrant of the buttock was held between two fingers of one hand; subsequently a needle (16–17 gauge), connected to a vacutainer system, was inserted with an angle of about 45° in the fat pad. After the insertion of the needle, the vacutainer tube was pressed forward to connect the vacuum with the needle. The needle was then carefully pushed back and forth 2–3 times within the fat pad to collect the adipose tissue biopsy. Subsequently the needle was immediately introduced in a sterile tube and frozen in liquid nitrogen. Adipose tissue was extracted from the needle with lysis solution (Purezool, Bio Rad, Milan, Italy) and then disrupted and homogenized using a tissue ruptor (Qiagen, Milan, Italy). Ribonucleic acid (RNA) was isolated using Aurum Total RNA Mini kit (Bio Rad) and stored at −80 °C until use. RNA labeling and hybridization on microRNA microarray chips was performed. Microarray results were analyzed using GeneSpring GX software 7.3 (Agilent Technologies). Data files were pre-processed using the GeneSpring plug-in for Agilent Feature Extraction software results. Data transformation was applied to set all the negative raw values at 5.0, followed by on-chip and on-gene median normalization. Filtering on-gene expression was applied so that probes expressed (flagged as Present) in at least one sample were kept and probes that did not change across all samples, identified as having a normalized expression always between median ± 1.5, were removed. Then, samples were grouped in accordance with their status and compared. Differentially expressed genes were selected as having a 1.5-fold difference between their geometrical mean expression, before and after BR [30].

### 2.7. Muscle Tissue Gene Expression

A fragment of the muscle biopsy sample was used to isolate total RNA, similarly to adipose tissue RNA extraction. cDNA was prepared from RNA using a High-Capacity cDNA Reverse Transcription Kit (Life Technologies Italia, Monza, Italy). Gene expression was measured by Real-Time PCR (StepOnePlus™, Applied Biosystems, Life Technologies Italia) using pre-designed TaqMan gene expression assays (Applied Biosystems, Life Technologies Italia: FNDC5, Hs00401 006 m1; Ribosomal Protein S13 (RPS13), Hs01 011 487 g1. The amplification reaction was performed in duplicate in 48-well plates. Expression data were normalized by the 2(**Δ**Ct) method using RPS13 as housekeeping gene and a reference sample to determinate fold increase of the target cDNA in the samples.

### 2.8. Statistics

Continuous variables were expressed as mean ± standard deviation (SD) or median (MD) and 95% confidence intervals (95% CI); categorical variables as frequencies. The continuous variables were first analyzed for normal distribution using Kolmogorov–Smirnov and Shapiro–Wilk tests. Variables not normally distributed were log transformed. At BDC, differences between groups and variable of interest were analyzed with variance analysis (ANOVA), with Bonferroni for post-hoc analysis, and chi-square test for categorical variables, while medians were compared by non-parametric tests (Kruskal–Wallis). Correlations between continuous variables were tested by Pearson’s correlation test for variables with normal distribution, whereas variables with non-normal distribution were analyzed after log transformation or with nonparametric test (Spearman’s test). The data are described with the correlation coefficient (r) and with the significance level against the null hypothesis (P_value_). For all the variables of interest, the variation, absolute (**Δ**) and percentage (**Δ**%), between BDC and BR14 was calculated (**Δ**variable = BR14 − BDC). We have also divided the subjects into tertiles of circulating levels of irisin at BDC and BR14 and into tertiles of the variation, absolute and percentage, of irisin between BDC and BR14 (tertiles **Δ**Irisin and tertiles **Δ**%Irisin).

Variations between BDC and BR14 in irisin plasma levels and the other variables of interest were analyzed by *t*-test for repeated measures and General Linear Model (GLM) Repeated Measures, Within-Subjects and Between-Subjects test.

Bivariate and multivariate (stepwise forward) linear regression analyses were followed to check the independence of the observed simple associations. Statistical analysis was performed using SPSS 26.0 software (SPSS Inc., Chicago, IL, USA) and statistical significance was set to P_value_ < 0.05. When significant a practical effect was also reported.

## 3. Results

Twenty-three male volunteers underwent a 14-day BR, seven young participants (mean age 23.3 ± 2.8 years) and sixteen older participants (mean age 59.3 ± 3.0). The muscle parameters and body composition of the entire population are shown in Appendix A. From BDC to the end of BR14, we observed changes of body composition (fat mass, FM, +1.4 kg (0.3–2.4 kg); fat free mass, FFM, −4.1 kg (−5.4–−2.7 kg)); muscle characteristic and performance (quadriceps femoris muscle volume, QMV, −133.9 cm^3^ (−165.8–−102.0 cm^3^); maximal explosive power of lower limb, MEP, −392.5 W (−514.4–−270.6 W); maximal voluntary contraction of knee extensors, MVC, −69.0 N (−100.4–−37.7 N)); and muscle fiber thickness and performance (cross-sectional area, CSA of Fiber Slow/1-791.7 m^2^ (−1808.7–225.3 µm^2^) and Fiber Fast/2-821.9 µm^2^ (−1654.4–10.7 µm^2^); isometric force, F_0_, of Fiber Slow/1-0.269 mN (−0.416–−0.122 mN) and Fiber Fast/2-0.219 mN (−0.392–−0.045 mN); specific force, P_0_ of Fiber Slow/1-44.5 mN mm^−2^ (−83.8–−5.3 mN mm^−2^) and Fiber Fast/2-20.1 mN mm^−2^ (−54.7–14.5 mN mm^−2^); unloaded shortening velocity, V_0_ of Fiber Slow/1-0.411 L s^−1^ (−0.611–−0.210 L s^−1^) and Fiber Fast/2-1.392 L s^−1^ (−1.897–−0.887 L s^−1^)) (all P_value_ < 0.005).

After BR, irisin concentration levels significantly increased (4.88 ± 2.72 mg/L vs. 6.76 ± 3.54 mg/L; P_value_ 0.003) (Figure 1A,B) despite showing great variability between different subjects. We did not find any association between baseline irisin and age (Appendix A) and any significant age group or cognitive training effect on post-BR irisin levels (Appendix A). Specifically, we performed correlation analyses showing no relationship between irisin, age or having performed CCT, or between these variables and absolute (**Δ**Irisin) or percentage (**Δ**%Irisin) changes in irisin levels from BDC to BR14 or between age-groups. Notably, both overall cohort and subgroup analyses provided similar results. Finally, given the great variability of irisin production, we divided the population first into tertiles with reference the basal irisin levels and then into tertiles with reference to the absolute (**Δ**Irisin) or percentage (**Δ**%Irisin) change of plasma irisin level during BR.

### 3.1. Baseline Correlations between Irisin and Body Composition, Muscle Performance, and Fiber Type Properties

BDC irisin concentration was associated to BMI (r, correlation coefficient; *r* 0.524; P_value_ 0.010) and absolute FFM (*r* 0.508; P_value_ 0.013), muscle mass (MM; *r* 0.556; P_value_ 0.006), and body cellular mass (BCM; *r* 0.547; P_value_ 0.007). At BR14, irisin concentration correlated positively with absolute and percentage fat mass (FM; *r* 0.519; P_value_ 0.011 and *r* 0.478; P_value_ 0.021, respectively) and negatively with percentage FFM (*r* −0.478; P_value_ 0.021). No statistical association was found, neither at BDC nor at BR14, between irisin concentration and muscle function (QMV; MEP; MVC) or muscle fibers type distribution or properties (CSA; F_0_; P_0_; V_0_) (Appendix A). We also evaluated the characteristics of Fiber Slow/1 and Fiber Fast/2 subclasses (Fiber Fast/2A, Fiber Fast/2A2X, Fiber Fast/2X) and found no correlations with irisin BDC and irisin BR14 concentration (data not shown).

### 3.2. Correlation between Irisin Variation and Variation of Body Composition, Muscle Performance, and Fiber Type Properties

No correlations were found between irisin at BR14 and absolute variations (**Δ**variable, difference between variable at BR14 vs. BDC) in the parameters of interest (Appendix A).

In contrast, percentage variation of irisin (**Δ**%Irisin) were correlated with several important parameters. Correlations of variations of 15 parameters (i.e., body composition, muscle performance, and muscle fibers characteristics) with variations of plasma irisin are shown in Appendix A while most significant association are shown in Figure 2. **Δ**Irisin was directly associated with **Δ**MVC, both as absolute value (*r* 0.476; P_value_ 0.022) and percentage (*r* 0.425; P_value_ 0.043) (Figure 2A, Appendix A). There were no direct correlations between **Δ**Irisin and other muscle performance parameters.

**Δ**Irisin was inversely associated with **Δ**CSA of Fiber Slow/1, both percentage (*r* −0.559; P_value_ 0.007) (Figure 2B) and absolute (*r* −0.614; P_value_ 0.002), and almost significantly negatively associated with absolute CSA variation of fiber type 2 (*r* −0.395; P_value_ 0.062) (Appendix A). Moreover, **Δ**%Irisin was associated with percentage of P_0_ variation of type fiber 1 (*r* 0.473; P_value_ 0.030) (Figure 2C). There were no significant associations between **Δ**Irisin and F_0_ and V_0_. We have also evaluated the variation of Fiber Fast/2 subclasses (Fiber Fast/2A, Fiber Fast/2A2X, Fiber Fast/2X) and we have not observed correlations with **Δ**Irisin (data not shown).

### 3.3. Effect of 14-Day BR and Tertiles BDC Irisin on Body Composition, Muscle Parameters, and Fiber Type Properties

To further investigate the impact of different amounts of irisin prior to intervention on functional and structural muscle parameters, we identified BDC irisin tertiles (1st 2.35–3.94; 2nd 4.00–4.66; 3rd 5.04–15.98 mg/dL) and evaluated whether the variations of a variable observed during BR were correlated by irisin tertiles.

We observed a negative BR effect on body composition (BMI, FFM, FM, BCM, MM) and muscle parameters (QMV, MEP, MVC), without BR/tertiles interaction effects (Appendix A).

Similarly, we observed a negative BR effect on CSA, F_0_, P_0_, V_0_; again, without BR/tertiles interaction effects (Appendix A). Moreover, we did not observe any effect on characteristics of the sub-classes of Fiber Fast/2 (Fiber Fast/2A, Fiber Fast/2A2X, Fiber Fast/2X) nor on the of these fibers (CSA, F_0_, P_0_, V_0_) (data not shown).

### 3.4. Effect of 14-Day BR and Tertile Irisin Variation on the Body Composition, Muscle Parameters, and Fiber Type Properties

To further evaluate the impact of plasma irisin concentration variation in different subjects, we also divided participants according to tertiles of Δ%Irisin (1st tertile −42.57–2.39; 2nd tertile 8.46–50.46; 3rd tertile 54.78–173.50%) (Appendix A). We observed a significant tertile effect, with the lowest decline of MVC (N) during BR in the highest tertile of Δ%Irisin as compared to the lowest tertile (−19.4 vs. −112.5 N; P_value_ 0.026); while only a trend in the tertile effect on specific MVC (N/cm^3^) was observed (−0.01 vs. −0.04 N/cm^3^; P_value_ 0.079) (Figure 3A,B).

Reduction in CSA of fiber 1 at BR14 showed a significant tertile effect in the highest tertile of Δ%Irisin, showing larger CSA reduction (−2993.5 vs. 405.4 μm^2^; P_value_ 0.003), while effect on fibers 2 CSA reduction was only close to reaching statistical significance (−2086.2 vs. −27.6 μm^2^; P_value_ 0.061) (Figure 3C,D).

We did not find a significant tertile Δ%irisin × BR interaction on observed reduction of P_0_ Fiber Slow/1. However, when we performed a *t*-test for repeated measures, we observed a reduction in P_0_ Fiber Slow/1 in the 1st and 2nd tertiles (P_value_ 0.051 and 0.10, respectively), while only the 3rd tertile did not change, suggesting that the effect could be significant with an increased statistical power. No other significant tertile Δ%irisin x BR interactions were observed with other fiber types properties (Appendix A). No significant changes were evident in fiber types distribution.

### 3.5. Bivariate and Multivariate Linear Regression Analyses

To evaluate whether irisin was a predictor of muscle parameters and fiber type properties, we performed bivariate linear regressions in which the independent variable **Δ**%Irisin was found significantly associated with 18% of **Δ**%MVC, 35% of **Δ**%CSA Fiber Slow/1, and the 22% of **Δ**%P_0_ Fiber Slow/1 (Appendix A).

Similarly, to assess whether and which muscle/fiber variables predicted **Δ**%Irisin, we carried out a multivariate stepwise-forward linear regression analysis in which, the dependent variable **Δ**%Irisin, was explained for 30% by the **Δ**%CSA Fiber Slow/1, independent of **Δ**%MCV and **Δ**%P_0_ of Fiber Slow/1 (Appendix A).

### 3.6. Tertiles of Δ%Irisin, Subcutaneous Adipose Tissue and Muscle FNDC5 Gene Expression

We found a reduction of FNDC5 gene expression in muscle tissue after BR in all Δ%Irisin tertiles, although this reduction is not statistically significant (Figure 4A). Conversely, we found that subjects in the highest tertile of Δ%Irisin showed the highest increase in FNDC5 gene expression in adipose tissue compared to baseline (1st and 2nd Tertile 1.4-fold change, 3rd Tertile 2.3-fold change, p Tertile 1st vs. 3rd 0.041; p Tertile 2nd vs. 3rd 0.038) (Figure 4B).

## 4. Discussion

To the best of our knowledge, this is the first time that irisin levels were evaluated after extreme physical inactivity rather than after exercise. We studied irisin plasma level in a 14-day horizontal BR protocol and analyzed relations between irisin and body composition, muscle function and composition and muscle fiber properties. We also studied FDNC5 gene expression in adipose and muscle tissues.

The role of age on circulating irisin levels is uncertain due to conflicting data [2,23,24]. We observed a lack of correlation between age, levels of irisin at BDC and BR14, and with the variation of irisin between BDC and BR14 (ΔIrisin), as well as no difference between groups (young vs. older) of irisin at BDC, BR14 and with Δirisin. Furthermore, no correlations were detectable between irisin and CCT. Thus, we analyzed the overall cohort to increase the statistical power of the analysis, confident that the same trend was observed in the analyses by age groups.

As previously reported [26], muscle mass, muscle volume, and muscle performance were all reduced post-BR, in parallel with a reduction of mean CSA, tension and velocity of all muscle fiber types [27].

Baseline irisin was associated to BMI, FFM, and BCM, while post-BR irisin was associated to FM. There are conflicting data about the relation of irisin and body composition: some authors observed a relation with BMI, FFM, and BCM [2] while others a direct correlation with FM [31]. Notably, in our study we observed changes in the pattern of association between irisin and post-BR body composition. The skeletal muscle is not the only source of irisin, since other tissues may contribute to its circulating levels [2]. Particularly, up to 28% of circulating irisin may arise from adipose tissue and 72% from muscles. The muscle/adipose irisin secretion ratio might vary; circulating irisin levels determined after exercise are strongly affected by muscle tissue, while the relative contribution of adipose tissue is increased in obesity [32]. Finally, in non-diabetic obese subjects, an increase in irisin production, both in muscle and adipose tissue, was observed as compared to lean subjects [20]. Accordingly, the positive association between baseline irisin concentration and FFM, and FM after BR, may reflect the reduction of FFM during physical inactivity. Therefore, the relative contribution of adipose tissue to circulating irisin may be increased, resulting to the positive correlation between irisin and FM, and the negative correlation with FFM after BR. In fact, we observed that the subjects with the highest increase in post-BR irisin concentration also had the greatest up-regulation of adipose tissue FDCN5 gene expression; conversely, post-BR FNDC5 gene expression was reduced in muscle tissue, regardless of circulating irisin variations.

In fact, in our study, circulating irisin surprisingly increased after BR despite being mainly characterized as an exercise-induced hormone. Moreover, irisin increase was inversely associated with the decrease of MVC and of P_0_ Fiber Slow/1. Other authors found a positive association between irisin and MVC [20]. When we divided the participants in tertiles of baseline irisin, we could not find any association. However, when we divided the participants in tertiles of irisin variation, we observed that subjects in the higher tertile of irisin increase had the lowest MVC and specific MVC reduction. These findings suggest that it is the amount of irisin produced during the BR that mediates a protective effect on MVC loss, more than the amount of irisin in circulation at the baseline. Intriguingly, the highest tertile of irisin variation showed a larger reduction of CSA, both in slow and fast fibers, without a clear group effect in F_0_ or P_0_. However, in the highest tertile of irisin variation, P_0_ of fiber type 1 did not change after BR, while it was reduced in the lowest tertile of irisin variation. This could suggest that muscle fibers of subjects in the highest tertile of irisin variation were the only ones able to maintain their own specific tension despite the major reduction of CSA. Subjects with the highest tertile of irisin variation were able to maintain fiber tension decline through a proportional reduction of CSA, an index of the amount of contractile myofibrils. Similarly, those individuals with the highest tertile of irisin variation experienced a less evident loss of MVC than other tertiles, despite a similar reduction in absolute muscle fibers tension; this suggests that there might be a mechanism that allows the muscle to exert a more MVC despite the same amount of fiber tension. In fact, it is well known that muscle fibers CSA directly affect muscle force but there are a lot of conditions that can modulate force independently of CSA, such as changes in muscle architecture, changes in the agonist/antagonist activation pattern, changes in the recruitment pattern and/or in the neural drive, or fiber type modifications [33]. Of course, all these factors may occur during BR [27]. Irisin produced during muscle exercise is involved in many pathways, finalized to increase muscle and metabolic capacity. We hypothesize that irisin production during inactivity might counterbalance the effect of disuse. Specifically, the disuse muscle atrophy may lead to an expansion of adipose tissue and induce adipose tissue to increase its own irisin production, mitigating the loss of muscle performance. In fact, we observed during BR a reduction of FNDC5 gene expression in skeletal muscle of all subjects with a parallel increase of its expression in adipose tissue in the highest tertile of irisin variation. In fact, those who increased the expression of FNDC5 in adipose tissue were those who increased the most the circulating irisin levels and were less affected by the adverse effects of muscle disuse. Thus, irisin might be a part of a complex muscle–adipose tissue crosstalk, through which the two tissues may influence each other, trying to re-establish a balance when a pathologic trigger perturbs the system. In this scenario, irisin production in adipose tissue might increase when there is an alteration of body composition with an absolute increase of fat mass, such as obesity, to counterbalance the relative deficiency of muscle. In the same way, irisin production in adipose tissue might increase during BR, in response to reduction of muscle activity and mass. The magnitude of the compensatory effect of irisin on muscle seems dependent on the amount of irisin produced more than the baseline irisin level. In fact, in animal studies, the infusion of high doses of recombinant irisin is able to attenuate muscle atrophy, due to mechanical unloading, showing less reduction in muscle mass and muscle CSA [8]. Animal studies used supra-physiological doses of irisin resulting in a supra-physiological activation of this pathway. Body irisin’s secretory capacity is limited. So, it is not surprising that we did not observe the restoration of muscle mass, but rather evidence of preserving deterioration of muscular performance.

The main strengths of this study are (i) we tested the bed rest, extreme physical inactivity intervention, as modifier of the muscle-adipose tissue crosstalk; (ii) we reported the effect of BR using a wealth of functional and morphological characteristics of the muscle, both macroscopic and microscopic, as well as evaluation of gene expression of both adipose tissue and muscle; and (iii) we could reveal the tissue source of the highest tertile of circulating irisin variation after BR. However, our findings should be interpreted in the light of several limitations. First, the small sample size of the study may have affected the generalizability of the results. However, the complexity of the study design and the intervention proposed has limited the number of participants to enroll. Second, the gene expression of adipose tissue varies in different body parts [34,35] so we cannot exclude that gluteal adipose tissue FNDC5 gene expression is similar to other adipose tissue sites. Third, the molecular mechanisms underlying the supposed adaptations are still not completely understood, yet the study design is not suitable for clarifying in more depth such aspects.

## 5. Conclusions

In conclusion, a reduction of fat free mass, muscle mass, muscle volume, and cross-sectional area of muscle fibers occurred after an experimental 14-day BR. Moreover, a decrease in contractile performance both at fiber and whole muscle level was evident. Unexpectedly, we observed a post-BR increase in the circulating levels of irisin with an inverse correlation between irisin change and CSA variation, especially of the Fiber Slow/1, without specific tension reduction; moreover, we observed a direct relationship between irisin variation and MVC. During a BR intervention, adipose tissue could substantially contribute to increase irisin concentration, as suggested by the increase of FNDC5 gene expression. We propose that participants who can increase levels of irisin lose less MVC and, therefore, better withstand the stress related to the BR. Further research is warranted to investigate the mechanisms underlying the effect of irisin on muscle during disuse, as the muscle-adipose tissue crosstalk might be a novel target to intervene on for improving the health of individuals with sarcopenia and muscle disuse atrophy.

## Figures and Tables

**Figure 1 biology-11-00999-f001:**
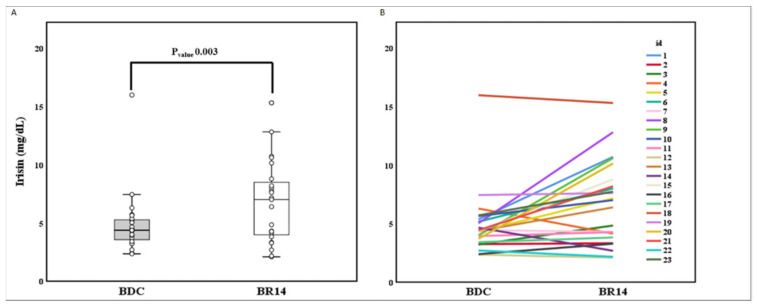
Effect of 14-day Bed Rest on irisin. (**A**): *t*-test pairs between BDC and BR14 irisin. (**B**): Spaghetti plot of irisin trend between BDC and BR14. BDC, baseline data collection; BR14, after 14-day Bed Rest data collection.

**Figure 2 biology-11-00999-f002:**
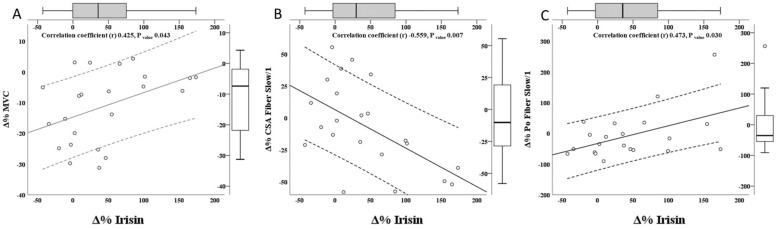
Correlation analysis between Irisin variation (percentage difference of the variables between BR14 and BDC) and fiber types properties variation (percentage difference of the variables between BR14 and BDC). In the different panels are described correlations between **Δ**Irisin (mg/L) BR14 vs. BDC (%) and (**A**): **Δ**MVC (%); (**B**): **Δ**CSA Fiber Slow/1(%); (**C**): **Δ**P_0_ Fiber Slow/1. BDC, baseline data collection; BR14, after 14-day Bed Rest data collection; **Δ**% variable, percentage difference of the variables of interest between BR14 and BDC; MVC, maximal voluntary contraction; CSA, cross-sectional area; P_0_, specific force.

**Figure 3 biology-11-00999-f003:**
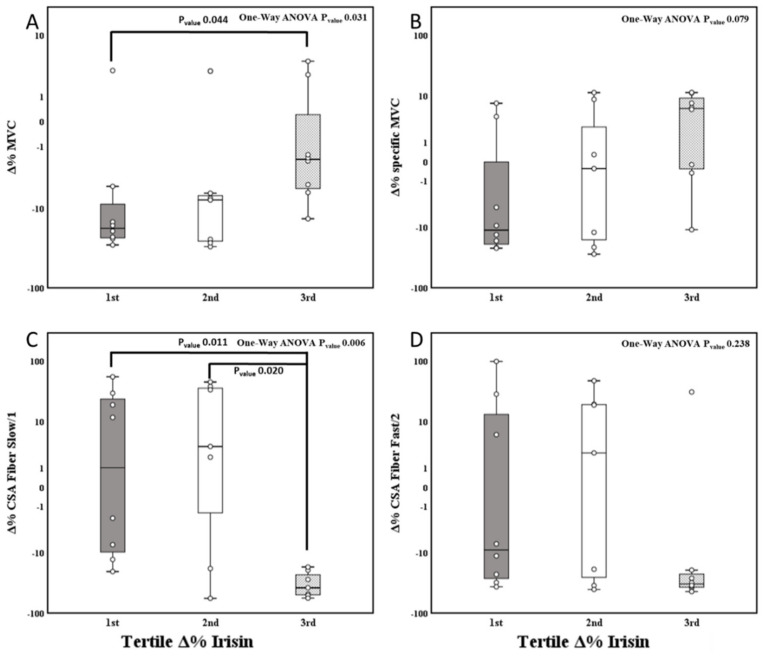
One-way analysis of variance for tertile irisin variation (percentage difference of the variables between BR14 and BDC) and muscle performance variation (percentage difference of the variables between BR14 and BDC). In the different panels are described for tertile **Δ**Irisin (mg/L) BR14 vs. BDC (%) the means (95% CI) of (**A**): **Δ**MVC(%); (**B**): **Δ**specific MVC(%). (**C**): **Δ**CSA Fiber Slow/1(%); panel (**D**): **Δ**CSA Fiber Fast/2. BDC, baseline data collection; BR14, after 14-day Bed Rest data collection; **Δ**% variable, percentage difference of the variables of interest between BR14 and BDC; MVC, maximal voluntary contraction; Specific MVC, ratio between MVC and QMV; CSA, cross-sectional area.

**Figure 4 biology-11-00999-f004:**
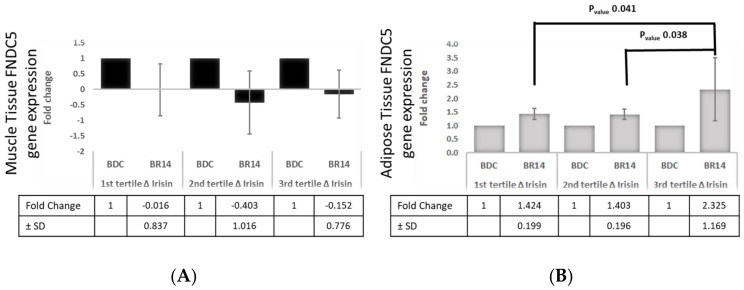
Effect of 14-day Bed Rest on the change of FNDC5 gene expression in muscle (**A**) and adipose tissue (**B**). BDC, baseline data collection; BR14, after 14-day Bed Rest data collection.

## Data Availability

The data supporting the study findings are available on request from the corresponding author [E.D.N. and A.P.]. Data are not publicly available, due to the PANGeA Study consortium agreement, which regulates the intellectual property of the data.

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
