# Peer review of "Irisin Attenuates Muscle Impairment during Bed Rest through Muscle-Adipose Tissue Crosstalk"

_biology, 2022, doi:10.3390/biology11070999_

Round 1

Reviewer 1 Report

Dear Authors

This manuscript gave us new information about the functional role of irisin between adipose and muscle in bed rest. 

Here are minor comments

1. Diagram Abstract about irisin, Figure 2 and Figure 3 should be enlarged although the page is increased. 

2. Font size in Tables should be increased. 

4. English should be improved and rephrased by Editing service. Some sentences are hard to read and understand. 

(1) therefore that there is a negative feedback mechanism within the muscle-adipose tissue crosstalk with which the muscle influences through irisin the adipose tissue during exercise but 39 also the adipose tissue protects the muscle from disuse during inactivity.

(2)  Detrimental effect of inactivity are well known. 

About what? and omitted "The"

(3) This study evaluates (in human), relationships between prolonged inactivity and circulating irisin, muscle performance, muscle fibers characteristics and FNDC5-gene expression (FNDC5ge) in muscle and adipose tissue.

and so on.

You needed to find out more sentences and rephrased them.   

Author Response

Dear Authors

This manuscript gave us new information about the functional role of irisin between adipose and muscle in bed rest.

Here are minor comments

  1. Diagram Abstract about irisin, Figure 2 and Figure 3 should be enlarged although the page is increased.

REPLY: We amended the diagram abstract and simplified Figure 2 and 3 as suggested

  1. Font size in Tables should be increased.

REPLY: Amended as suggested

  1. English should be improved and rephrased by Editing service. Some sentences are hard to read and understand.

REPLY: The manuscript has been revised to improve the quality of English form. 

Reviewer 2 Report

The Authors indicated an increase in FNDC5 gene expression in subcutaneous adipose tissue and not in skeletal muscle after 14 days-bed rest of healthy voluntaries. This peculiar finding that link highest increase of serum irisin during inactivity is really intriguing and new in the panorama of irisin production and skeletal muscle but must be better organized and supported by more focused text.

However, in my opinion many criticisms are made to revise the study as follows:

Major CHANGES

1.Firstly, subcutaneous gluteal tissue analysed in this study has been aspirated by the gluteal area. This peculiar tissue has been reported to be significantly different from human neck subcutaneous white and abdominal visceral fat. This characteristic is further accentuated in female vs male. For this reason, sex dependent effect of bed resting must be better monitored. See Karastergiou et al J Clin Endocrinol Metab 98, 362-371,2013; Karastergiou et al Adv Exp Med Biol 2017.

2.FNDC5 increase obtained in the adipose tissue and not in skeletal muscle has been inserted at the end of the Results section but this is the most important finding as indicated also in the title.

3 BDNF levels during bed rest have been hypothesized to be due to FNDC5/irisin in adipose tissue even if the authors did not show any change. Please discuss better this point or delete the sentences from line 468 to 473.

4.The Authors might resume and better reorganize statistically not significant data inserted in the text. Too many informations make hard to understand the novelty of this study as stated above.

5.I suggest reconsidering the Figures and Legends and eventually to insert data as Supplementary files for Figure 2 and Figure 3.

Minor CHANGES

1.Please write abbreviations in extenso in an Abbreviations List and put it at the end of the main text.

2.Please enlarge plots that are hardly understandable and chose more representative one. Figure 2, Table 1, Table 2, Table 3, Table 4.

3. Rewrite Figure 2 and Figure 3 Legends and reduce the number of abbreviations.

Author Response

The Authors indicated an increase in FNDC5 gene expression in subcutaneous adipose tissue and not in skeletal muscle after 14 days-bed rest of healthy voluntaries. This peculiar finding that link highest increase of serum irisin during inactivity is really intriguing and new in the panorama of irisin production and skeletal muscle but must be better organized and supported by more focused text.

REPLY: We thank the reviewer for the positive feedback, and we have addressed in the revised version the issues raised.

However, in my opinion many criticisms are made to revise the study as follows:

Major CHANGES

1.Firstly, subcutaneous gluteal tissue analysed in this study has been aspirated by the gluteal area. This peculiar tissue has been reported to be significantly different from human neck subcutaneous white and abdominal visceral fat. This characteristic is further accentuated in female vs male. For this reason, sex dependent effect of bed resting must be better monitored. See Karastergiou et al J Clin Endocrinol Metab 98, 362-371,2013; Karastergiou et al Adv Exp Med Biol 2017.

REPLY: We thank the reviewer for this comment. We recognized that there are substantial differences across different sites of the adipose tissue.  In fact, our group has also previously published a paper on this specific topic on the same cohort of healthy individuals (Passaro, Angelina, et al. "Gene expression regional differences in human subcutaneous adipose tissue." BMC genomics 18.1 (2017): 1-11.). Unfortunately, after the genetic investigations carried out, we could not have enough biological samples  to proceed with the evaluation of the gene expression of FND5C from different adipose tissue sites.Therefore we have added a specific comment on this issue in the limitation section of the discussion (Page 12 Lines 464-465)

2.FNDC5 increase obtained in the adipose tissue and not in skeletal muscle has been inserted at the end of the Results section but this is the most important finding as indicated also in the title.

REPLY: We thank the reviewer for this comment and we agreethat the FNDC5 gene expression are among the major finding of the study relevance., Nevertheless, we think that  it  is instrumental for a better understanding of the readers to report the findings starting from the description at body level (i.e., body composition), then the muscle macroscopically (i.e., volume and strength of the muscle) and microscopically (i.e., properties of the fibers) followed bythe current relationship between these characteristics and irisin’s behavior. Lastly, to keep the flow of the manuscript consistent with our reasoning during the study we reported the data on the investigation of the likely tissue source of irisin. . That being said, whether the Editor and the Reviewer think that this is a deal-breaker for the acceptance, we are available for restructuring the results section as suggested

3 BDNF levels during bed rest have been hypothesized to be due to FNDC5/irisin in adipose tissue even if the authors did not show any change. Please discuss better this point or delete the sentences from line 468 to 473.

REPLY: We thank the reviewer for this comment. We agree with the suggestion and deleted the highlighted paragraph until more data will be available on this specific topic.

4.The Authors might resume and better reorganize statistically not significant data inserted in the text. Too many informations make hard to understand the novelty of this study as stated above.

REPLY: We thank the reviewer for the comment. We believe it is important to report also the negative data in order to have the most detailed and reliable characterization of the observed phenomena. In order to improve the results section and highlight the novelty of the study, we have implemented the figures showing the most relevant findings and keep the non-significant findings only in the text.  "

  1. I suggest reconsidering the Figures and Legends and eventually to insert data as Supplementary files for Figure 2 and Figure 3.

REPLY: We thank the reviewer for the suggestion. We agree that the paper is particularly rich in information that burdens the reading.  Nevertheless, we believe that reporting detailed findings better inform the understanding of the readers on the complexity of the observed phenomena. . We simplified the results section moving in supplemental materials part of the correlation analyses and kept in the main text the most relevant data.

Minor CHANGES

1.Please write abbreviations in extenso in an Abbreviations List and put it at the end of the main text.

REPLY: We have included at the end of the main text the abbreviations list as suggested.

2.Please enlarge plots that are hardly understandable and chose more representative one. Figure 2, Table 1, Table 2, Table 3, Table 4.

REPLY: We agree with the suggestion. We have enlarged the plots and chosen the most representative ones. Therefore, we moved to the supplemental material some tables and figures.

  1. Rewrite Figure 2 and Figure 3 Legends and reduce the number of abbreviations.

REPLY: Amended as suggested.

Reviewer 3 Report

In "Irisin attenuates muscle impairment during bed rest through 2 muscle-adipose tissue crosstalk" the authors present findings for a 14 day bedrest study. In this study only 23 males were studied and there were 7 young and 16 older individuals. The general characterization suggests the bedrest intervention worked as similar to previous interventions. The main novel finding is examination of Irisin levels in response to bedrest and with respect to production in muscle vs. fat. The graphical abstract nicely conveys this.

Given this is the main finding there are allot of extra "negative" results presented. These analyses could be removed to make the paper easier for the reader (I also understand why they are presented). Removal of the negative data, in my opinion, would not detract from the overall impact, interest, and quality of the work. 

The one piece of data not presented (which may or may not be negative data) is the effect of age (n=7 vs. n=16) on the overall study and, particularly, Irisin results. Inclusion of this if not negative could improve the paper. Inclusion of this if negative or not sufficiently powered, however, does not seem a good plan.

Overall, the paper could do with some further editing but the results speak for themselves and the reason for individual variation remains interesting as well for future work.

Author Response

In "Irisin attenuates muscle impairment during bed rest through 2 muscle-adipose tissue crosstalk" the authors present findings for a 14-day bedrest study. In this study only 23 males were studied and there were 7 young and 16 older individuals. The general characterization suggests the bedrest intervention worked as similar to previous interventions. The main novel finding is examination of Irisin levels in response to bedrest and with respect to production in muscle vs. fat. The graphical abstract nicely conveys this.

Given this is the main finding there are allot of extra "negative" results presented. These analyses could be removed to make the paper easier for the reader (I also understand why they are presented). Removal of the negative data, in my opinion, would not detract from the overall impact, interest, and quality of the work. 

REPLY: We thank the reviewer for the feedback and the valuable suggestions. We agree that the reported negative results might make the reading slower and more difficult, however we also believe that the non-significance of some findings is also relevant to follow the flow of our research approach. To simplify the main text we have restructured and moved to the supplemental materials some tables and figures. We kept in the main text only the more impactful images We hope that these changes will contribute to improve the clarity of the manuscript.

The one piece of data not presented (which may or may not be negative data) is the effect of age (n=7 vs. n=16) on the overall study and, particularly, Irisin results. Inclusion of this if not negative could improve the paper. Inclusion of this if negative or not sufficiently powered, however, does not seem a good plan.

REPLY: We thank the reviewer for this comment. Now the the information requested is reported in the supplementary materials (Appendix A; Tables S2-S3)

Overall, the paper could do with some further editing but the results speak for themselves and the reason for individual variation remains interesting as well for future work.

REPLY: We sincerely hope that the reviewer will find the revised version of the manuscript improved and suitable for publication.

Round 2

Reviewer 2 Report

The authors answered properly to many criticisms and followed referee' suggestions. This revised version may be accepted, even if minor editing changes are performed on figures and legends.